# CFD Simulation Based Investigation of Cavitation Dynamics during High Intensity Ultrasonic Treatment of A356

Eric Riedel *, Niklas Bergedieck and Stefan Scharf

Institute of Manufacturing Technology and Quality Management, Otto-von-Guericke-University Magdeburg, Universitätsplatz 2, 39106 Magdeburg, Germany; niklas.bergedieck@ovgu.de (N.B.); stefan.scharf@ovgu.de (S.S.)
*   Correspondence: eric.riedel@ovgu.de; Tel.: +49-391-67-57084

**Abstract:** Ultrasonic treatment (UST) and its effects, primarily cavitation and acoustic streaming, are useful for a high range of industrial applications, e.g., welding, filtering, cleaning or emulsification. In the metallurgy and foundry industry, UST can be used to modify a material's microstructure by treating metal in the liquid or semi-solid state. Cavitation (formation, pulsating growth and implosion of tiny bubbles) and its shock waves, released during the implosion of the cavitation bubbles, are able to break forming structures and thus refine them. In this context, especially aluminium alloys are in the focus of the investigations. Aluminium alloys, e.g., A356, have a significantly wide range of industrial applications in automotive, aerospace and machine engineering, and UST is an effective and comparatively clean technology for its treatment. In recent years, the efforts for simulating the complex mechanisms of UST are increasing, and approaches for computing the complex cavitation dynamics below the radiator during high intensity ultrasonic treatment have come up. In this study, the capabilities of the established CFD simulation tool FLOW-3D to simulate the formation and dynamics of acoustic cavitation in aluminium A356 are investigated. The achieved results demonstrate the basic capability of the software to calculate the above-mentioned effects. Thus, the investigated software provides a solid basis for further development and integration of numerical models into an established software environment and could promote the integration of the simulation of UST in industry.

**Keywords:** aluminium; ultrasonic melt treatment; cavitation; CFD simulation; structure refinement

## 1. Introduction

Cavitation, caused by high-intensity ultrasonic treatment (UST), is used in a wide range of industrial applications and becomes more and more interesting for metallurgy and foundry processes. It can be used as an effective method for modifying a material's microstructure and to improve the material's mechanical properties [1,2]. Especially in the context of the treatment of aluminium alloys, many investigations were conducted in the last 20 years. Aluminium and its alloys are thereby of primary importance as they are used for the manufacturing of a wide variety of industrial components in automotive, aerospace, shipping and mechanical engineering. The UST of aluminium can be conducted in a liquid or semi-solid state [3–9]. In the process, extreme pressure amplitudes (higher than the material's cavitation threshold) are induced by an immersed sonotrode through high-frequency sinusoidal mechanical vibration, which causes the formation of cavitation bubbles during the phase of negative pressure amplitudes, pulsating growth and the collapse of cavities accompanied by energy release in the form of shock waves [1,10–12]. Since the

pressure intensity decreases exponentially with growing distance, in most cases, cavitation activity is limited to a small cavitation cloud close to the sonotrodes tip [1,13]. The mentioned dynamics within the cavitation cloud, including temporal pressure and temperature changes, can evoke several refining metallurgical effects, mainly:

- Wetting: The aluminium melt inevitably contains low amounts of $Al_2O_3$, a by-product resulting from the hydrogen-absorbing reaction of aluminium and $H_2O$ vapour. On the surfaces of these $Al_2O_3$ particles, hydrogen deposits prevent $Al_2O_3$ from wetting. Shock waves, resulting from the collapse of cavitation bubbles near the particle, remove the hydrogen deposits and make the $Al_2O_3$ particles available as nucleation sites for heterogeneous nucleation [9,14–16].
- Nucleation: The released shock waves result in a change of pressure ratios close to the collapsing bubbles. This leads to an increase of the alloy's solidification temperature, and thus to the formation of solid aluminium grains. Below liquidus temperature, a few of the grains are thermally stable and survive the drop to normal ambient pressure; they support microstructures' refinement as available nucleation sites [5,15–19].
- De-agglomeration: The releasing shock waves are able to separate agglomerated particles and, in this way, enhance the particle distribution and increase the amount of available nucleation sites within the scope of heterogeneous nucleation [1,16].
- Fragmentation: This effect takes place at temperatures below the liquidus temperature, when the alloy starts to solidify and dendrites form. Pulsating cavitation bubbles close to the dendrites and its roots bend the dendrite arms during pulsation regularly. Either during bending or through shock waves, dendrite arms break. The broken off dendrite arms henceforth are the basis for growing dendrite structures [1,16,20–26].

The knowledge of the development, size and dynamics of cavitation (zone) is of high importance for effective usage of the mentioned effects, e.g., for the process design of ultrasonic supported systems. In recent years, efforts for simulating ultrasonic treatment and its effects have increased. An overview of most of the published simulation studies so far was given in [27]. So far, much progress has been made, but looking at the efforts for simulating ultrasonic treatment in its entirety, it still is in the stage of development. While we presented a comprehensive approach for the simulation of UST in [27], the study lacked a detailed investigation of cavitation dynamics. Therefore, in this study, the capabilities of the established CFD (casting) simulation tool FLOW-3D, developed by Flow Science, Inc. (Santa Fe, NM, USA) to calculate and visualize cavitation in aluminium melt, here A356, were investigated.

## 2. Numerical Modelling

### 2.1. General

All modelling and simulation activities were performed using the CFD simulation software FLOW-3D v11.2, whereas for analysis purposes, the software FlowSight v11.2 was used. Both programs were developed by Flow Science, Inc. Within the scope of the investigations, a compromise had to be found between the resolution of the visual results (depending on the cell size) and the associated computing time (depending on the cell size, the time step definition and the CPU power). As a minimum requirement, the definition of the time steps should allow the most accurate reproduction of the sinusoidal curve on which the sonotrode movement is based (see Section 3.1). For this reason, the investigated time frame for the development of cavitation was set to 0.001 s, and the time step for calculation was set to $1.5625 \times 10^{-6}$ s.

## 2.2. Meshing and Geometry

To minimize the required simulation time, only necessary and essential elements were modelled and simulated, without neglecting a real system. The fluid volume was 8 × 8 × 8 mm (xyz), deducting the immersed sonotrode volume. The sonotrode used had a diameter $d_s$ of 5 mm and an immersion depth of 4 mm, and its material was defined as ceramic. The selected dimensions allowed for a very small cell size with an acceptable amount of cells. In addition, particularly in the case of in situ experiments, sonotrodes with diameters of just several mm are used [19,25,28,29]. Since the investigations were conducted in isothermal conditions, heat transfer could be neglected. Thus, no sonotrode temperature was defined. The radiator movement was controlled by a sinusoidal translational velocity component in the z-direction, corresponding to an ultrasonic system with 20 kHz and a peak-to-peak amplitude of 35 µm, which leads to 20 calculated oscillations. The atmospheric pressure and temperature were set to 101,325 Pa and 293.15 K, respectively. Due to the reasons mentioned above, the cell size was $5 \times 10^{-5}$ mm, and the primary three-dimensional system was discretised with 4,608,000 cells. That way, the corresponding visual results have a high resolution at acceptable computing times. To let acoustic waves propagate across the boundaries and avoid reflecting effects close to the radiator, multi-block meshing was defined. In this case, it means that the mesh block of the primary system (Mesh Block 1) is completely enclosed by another mesh block (Mesh Block 2) with a cell size of 1 mm. The boundaries of Mesh Block 1 were defined as inter-block mesh boundaries. Figure 1 displays the described modelling of Mesh Block 1.

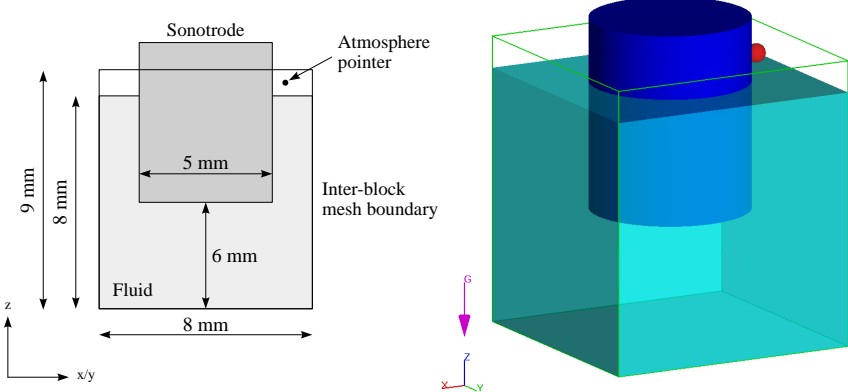

**Figure 1.** Geometric alignment of the simulation model setup.

## 2.3. Fluid

Fluid properties were set corresponding to aluminium silicon alloy A356 at 973.15 K, since the speed of sound within aluminium is known for this temperature from [30]. The main parameters used are presented in Table 1. To account for acoustic waves in fluids that would otherwise be treated as incompressible, the (limited) compressibility model was activated. That way, it is possible to simulate sharp pressure changes, like those occurring during UST. The compressibility coefficient $\beta$ is defined as:

$$\beta = \frac{1}{\rho c^2} \tag{1}$$

where $\rho$ and $c$ are the respective fluid density and adiabatic speed of sound [31]. Therefore, the material's density and speed of sound can be considered for the calculation of acoustic pressure.

| Parameter | A356 | Unit |
|---|---|---|
| Density | 2437 | $kg/m^3$ |
| Viscosity | 0.0019 | kg/m/s |
| Specific heat | 1074 | J/kg/K |
| Thermal conductivity | 86.9 | W/m/K |
| Liquidus temperature | 881.15 | K |
| Solidus temperature | 825.55 | K |
| Speed of sound | 4600 | m/s |
| Compressibility | 1.94 | 1/Pa |
| Surface tension | 0.871 | $kg/s^2$ |

*2.4. Cavitation Physics*

Within the cavitation model, cavitation is measured by a transport equation, which calculates the advection, production and dissipation of the cavitation volume fraction, according to the following equations, which are the default equations for FLOW-3D v11.2 [31]:

$$\frac{DV_{cav}}{D_t} = Cav_{production} - Cav_{dissipation}, \tag{2}$$

$$C_p = C_e \frac{E_{turb}}{\sigma} \rho_l \rho_v \sqrt{\left[\frac{2}{3}\frac{p_{cav}-p}{\rho_l}\right]} (1 - f_{cav}) \tag{3}$$

$$C_d = C_c \frac{E_{turb}}{\sigma} \rho_l^2 \sqrt{\left[\frac{2}{3}\frac{p-p_{cav}}{\rho_l}\right]} f_{cav} \tag{4}$$

$V_{cav}$ is the computed cavitation volume fraction; $C_p$ is the cavitation production coefficient; $C_d$ is the cavitation dissipation coefficient; $C_e$ the evaporation coefficient; $C_c$ the condensation coefficient; $E_{turb}$ is the turbulent kinetic energy (alternatively 10% of the total kinetic energy if no turbulence model is selected); $p_{cav}$ is the specified cavitation pressure; $p$ is the local fluid pressure; $\sigma$ is the material surface tension; and $f_{cav}$ is the mass fraction of cavitation within the cell, with $\rho_l$ and $\rho_v$ as the densities of liquid and vapour. The coefficients used for cavitation production and dissipation are default values (0.02 and 0.01, respectively) that can be adjusted. The newly opened void region was treated as a fixed pressure bubble with the pressure equal to the cavitation pressure, which is a fixed parameter that currently does not depend on temperature [31]. Hydrogen gas at 973.15 K was assumed to be the gas species inside the cavitation bubbles. The respective gas density was calculated by the general gas equation:

$$pV = nR_S T \longrightarrow \frac{n}{V} = \frac{p}{R_S T}, \tag{5}$$

where $p$ is the gas pressure, $V$ the gas volume, $n$ the amount of gas, $R_m$ the universal gas constant and T the thermodynamic temperature of the gas [32]. Using the molar mass of $H_2$ and the data in Table 2, the gas density of cavitation bubbles was 0.025 $kg/m^3$. While in real cases, it is expected that the pressure within the bubbles changes with positive and negative pressure amplitudes resulting from the radiator, the average value still is the atmospheric pressure. Due to simplification, the average value of hydrogen density was used within a so-called one fluid simulation to reduce the complexity of the simulation model in which one density within the bubbles can be defined. Within the scope of the bubble and phase change model, the relationship between pressure, volume and temperature followed an adiabatic law. The pressure in each bubble was inversely proportional to its volume to the power of $\gamma$. Since the bubble and phase change model allows tracking collapsed bubbles with void particles, this feature was activated

to analyse the distribution of cavitation activity within the analysed volume. For an accurate calculation and visualization, opening bubbles had to be resolved by a minimum of three cells across the diameter [31].

**Table 2.** Parameters and values used for the calculation of hydrogen density at 973 K [32].

| Parameter | Value | Unit |
|---|---|---|
| Gas pressure $p$ | 101,325 | Pa |
| Universal gas constant $R_m$ | 8314.41 | J/(kmol K) |
| Temperature T | 973.15 | K |
| Molar mass M ($H_2$) | 2.016 | kg/kmol |

The so-called general moving object (GMO) model calculates the radiator movement by a sinusoidal translational velocity component in the z-direction, corresponding to an ultrasonic system with 20 kHz and a peak-to-peak amplitude of 35 µm. The sinusoidal translational velocity is calculated by:

$$v(t) = \omega \cdot s_0 \cdot cos(\omega t + \phi_0), \tag{6}$$

with:

$$\omega = 2\pi f, \tag{7}$$

where $v(t)$ corresponds to the angular frequency, $\omega$ is the angular frequency with $f$ as the frequency, $s_0$ is the amplitude (17.5 µm) and $\phi_0$ is the phase angle. The GMO model calculates the kinetic energy, which is caused by the radiator movement, and its transfer on the fluid by momentum and mass conservation. For moving object/fluid coupling, the explicit numerical method was chosen, in which the fluid and GMO motions, i.e., the radiator, of each time step are calculated using force and velocity data from the previous time step. Besides gravitation and the activation of the GMO model for the movement of the radiator, the surface tension model, as well as the viscosity and turbulence model were of primary importance. All specific parameters and values used are listed in Table 3.

**Table 3.** Summary of the chosen models with the corresponding parameters.

| Model | Parameter | A356 | Unit |
|---|---|---|---|
| Bubble and phase change with adiabatic bubble and dynamic nucleation | Gamma | 1.4 | Without unit |
| | Pressure | 101,325 | Pa |
| Cavitation with empirical model for cavitation control | Cavitation pressure (Cavitation threshold) | 0 | Pa |
| active model for voids and | Surface tension coeff. | 0.871 | $kg/s^2$ |
| activated cavitation | Density of cav.bubbles | 0.025 | $kg/m^3$ |
| potential model | Cav. production coeff. | 0.02 | Without unit |
| | Cav. dissipation coeff. | 0.01 | Without unit |
| Surface tension model with | Surface tension coeff. | 0.871 | $kg/s^2$ |
| explicit numerical | Temperature dependence | 0 | $kg/s^2/K$ |
| approximation for surface tension pressure | Contact angle | 90 | Degrees |

## 3. Results

### 3.1. Radiator Movement

First, the software's capabilities to follow the sinusoidal curve shape and to calculate the entered amplitude were analysed. The result is demonstrated in Figure 2. The chosen time step to calculate the oscillation can be considered sufficient to correctly generate the targeted amplitude of 35 µm. Both calculated

amplitude and frequency correspond to the entered parameters and are not impaired by the chosen time steps. Therefore, the chosen time step can be considered a solid basis for the following investigations.

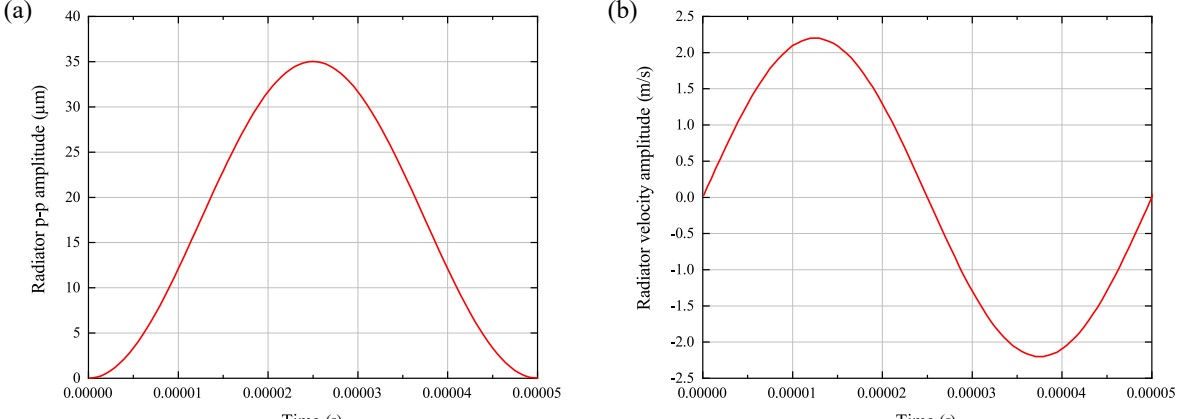

**Figure 2.** Analysis of calculated radiator movement: (**a**) actually calculated radiator amplitude and (**b**) velocity amplitude and frequency.

### 3.2. Cavitation

Figure 3 shows the juxtaposition of the cavitation volume fraction, pressure analysis and fluid fraction for five different times in the cells right below the radiator surface. The analysis of the cavitation volume fraction (Figure 3a) reveals a fast increase in cavitation activity within the first 0.0003 s. From there, the cavitation zone starts expanding in all directions in the form of little branches. The distribution of cavitation activity seems mostly homogeneous; only in some sectors, slightly stronger activities are measureable. The pressure analysis in Figure 3b) allows gaining better insight into the pressure conditions within the cavitation zone. It reveals the permanently varying pressure distribution within the cavitation zone and, in addition to it, the formation of locally occurring positive pressure hotspots, the values of which are much higher than the usual pressure amplitudes. Furthermore, the boundaries of the active cavitation zone towards the rest of the fluid are partially clearly recognisable. Volume fraction rendering (Figure 3c) highlights sectors with volume fraction values between 1.0 (100% fluid) and 0.0 (100% void). This way, e.g., the occurring bubbles or cavities are made visible, and better insight into the cavitation activity is possible. At t = 0.0001 s, a conspicuous ring structure is visible. Considering the radiator movement (Figure 2b), it becomes evident that the radiator reached its negative location peak at t = 0.0001 s. Therefore, it can be assumed that the ring structure arises from the fluid displacement caused by the radiator.

In addition to the cavitation volume fraction results shown in Figure 3a, the cavitation area dimensions in yz-layer in the cross-section of the sonotrode after t = 0.001 s are demonstrated in Figure 4. Starting from the radiator surface, the cavitation area also expands downwards. The cavitation area can thereby be divided in two zones: a primary zone (zone 1), where the cavitation activity is highest and a transition zone (zone 2) between zone 1 and the rest of the fluid, where the cavitation activity is continuingly decreasing to zero. Since the pressure amplitude decreases exponentially with growing distance to the radiator and cavitation dynamics are directly dependent on the pressure conditions, cavitation activity decreases as well.

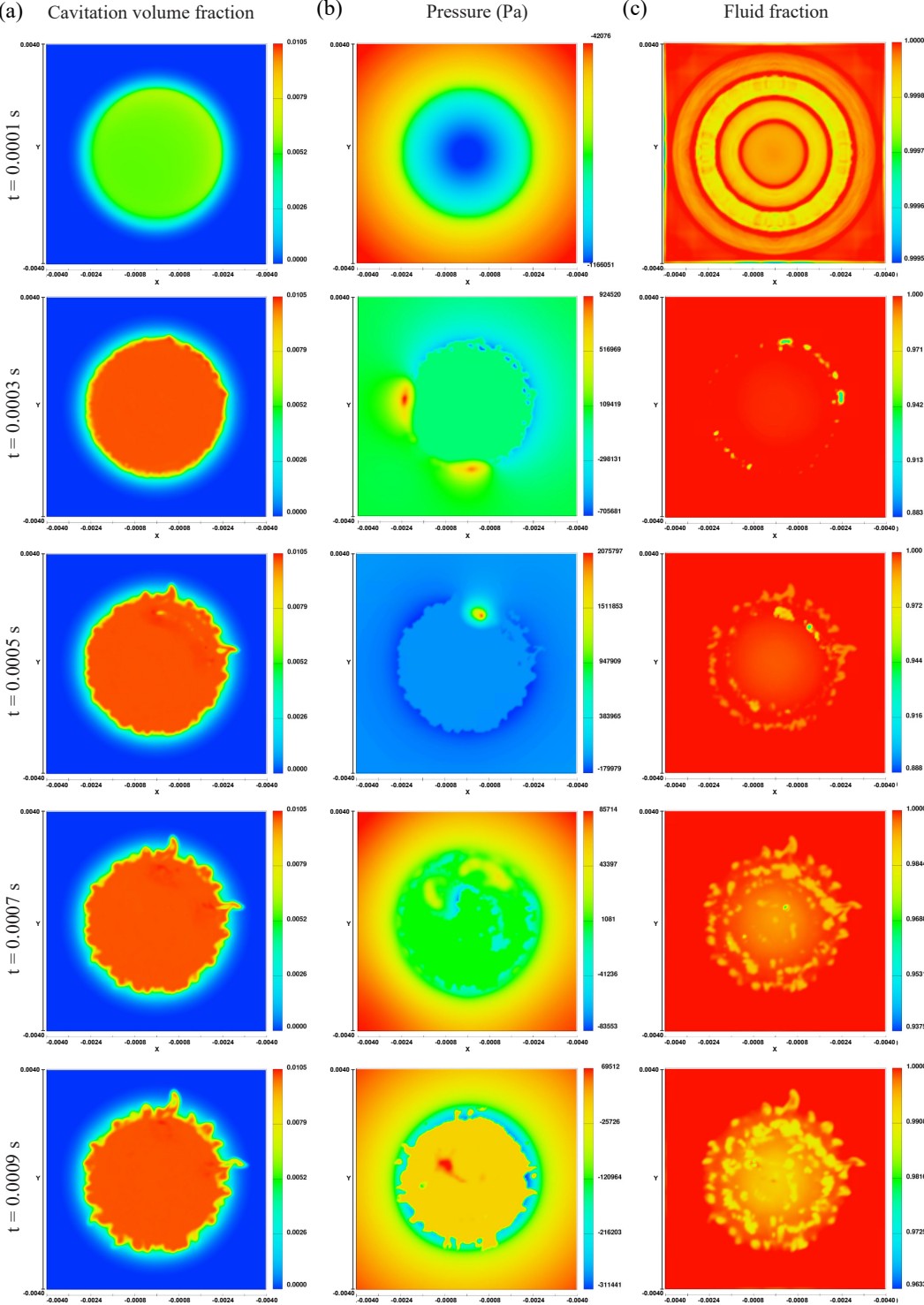

**Figure 3.** Development of the cavitation zone on the radiator's surface at different times: (**a**) cavitation volume fraction, (**b**) pressure and (**c**) fluid fraction.

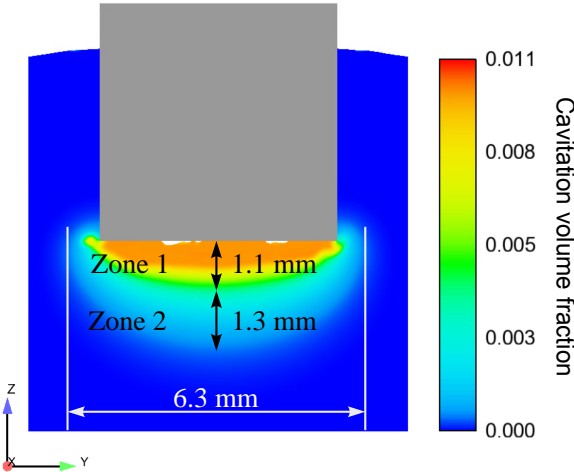

**Figure 4.** Dimensions of the cavitation zone in the central yz-layer after t = 0.001 s.

In addition to the results illustrated in Figure 3, the volume fraction distribution in and around emerging bubbles is calculated, as demonstrated in Figure 5. The displayed bubble in Figure 5a has a maximum size of four mesh cells, which corresponds to roughly 200 µm, and forms and collapses within 30 µs. Figure 5b shows the corresponding pressure values during bubbles' expansion and compression and after bubble collapse. Until the final collapse, the bubble is surrounded by a negative pressure field, which adjusts to the pressure values of the rest of the fluid with increasing distance to the bubble. After the bubble vanishes or collapses (t = 0.000240 s), a short pressure hotspot occurs.

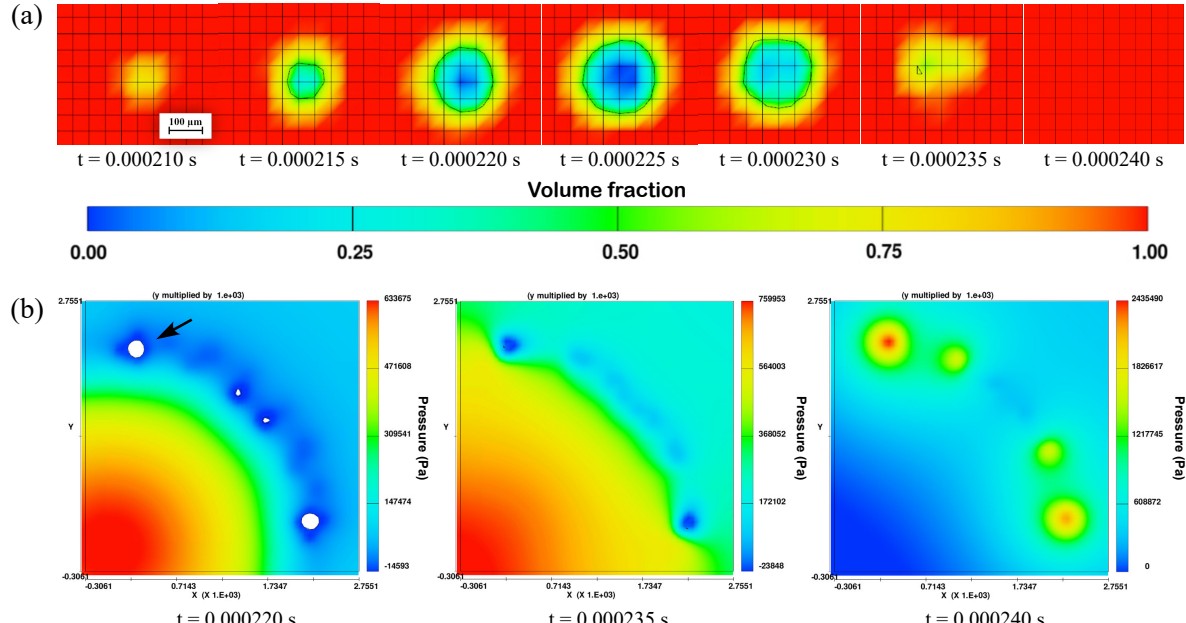

**Figure 5.** Bubble dynamics: (**a**) fluid fraction in and around the bubble and (**b**) pressure conditions in the area around the bubble and after collapse.

Apart from this, the volume fraction rendering shows that proceeding from the spherical radiator boundaries, more and more areas with volume fractions lower than 100% arise. From there, these areas increasingly appear right up to the radiator's surface centre with every oscillation and cover the whole radiator surface. Figures 6 and 7 support the observation that the occurring cavities seem to expand from the radiator surface edge to its centre. With every oscillation, the area without volume contact spreads in the direction of the surface centre. This occurs when the radiator is moving upwards, and a negative acoustic pressure builds up and disappears or collapses when the sonotrode changes its direction and a positive pressure amplitude builds up.

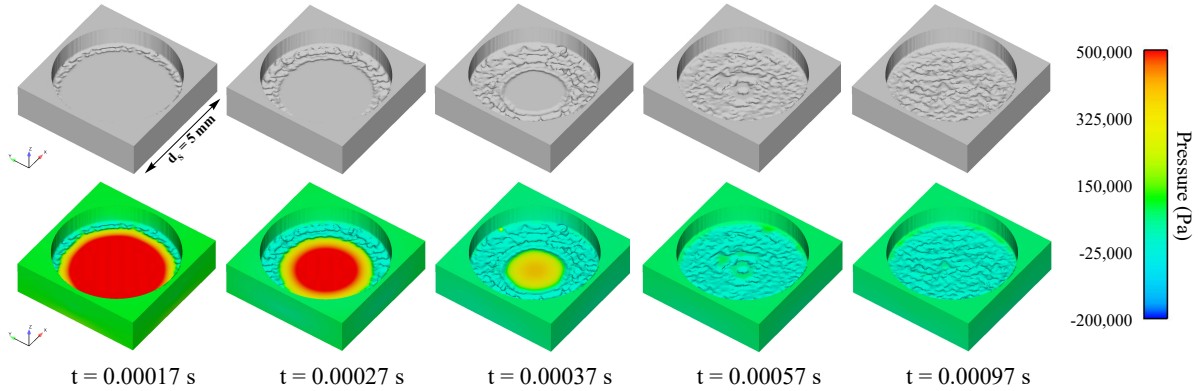

**Figure 6.** Expanding cavitation activity and respective pressure build-up.

### 3.2.1. Shielding Effect

Since it is known that cavitation influences the acoustic pressure propagation and that it is an important factor for cavitation dynamics, the pressure oscillation with and without the activated cavitation model was analysed. Figure 8 shows the comparison of the pressure developments in the centre line 4 mm below the radiator. Without the activated cavitation model, pressure oscillates sinusoidally around the ambient pressure of almost 100,000 Pa regularly, and no abnormalities are registered; whereas with the activated cavitation model, considerable fluctuations in pressure values are perceptible. It can be concluded that the cavitation activity has a damping influence on acoustic pressure development and propagation, even if pressure eruptions higher than the normal pressure oscillation (without cavitation) occur, due to collapsing bubbles, were measured.

### 3.2.2. Collapsing Bubbles

As already mentioned, the software is able to track collapsed bubbles via void particles as a quantitative measurement that fulfil the function of markers. Within the investigated time period (1000 µs), one-thousand seven-hundred fifty-two collapsed bubbles were registered. Figure 9 demonstrates the overlay of collapsing bubbles with the corresponding pressure values at that exact moment. It reveals that the collapse activity increases and decreases the more the positive pressure amplitude nears and leaves its local peak. Nearly all collapses take place close to the radiator surface; collapsing activity decreases with increasing distance to the radiator.

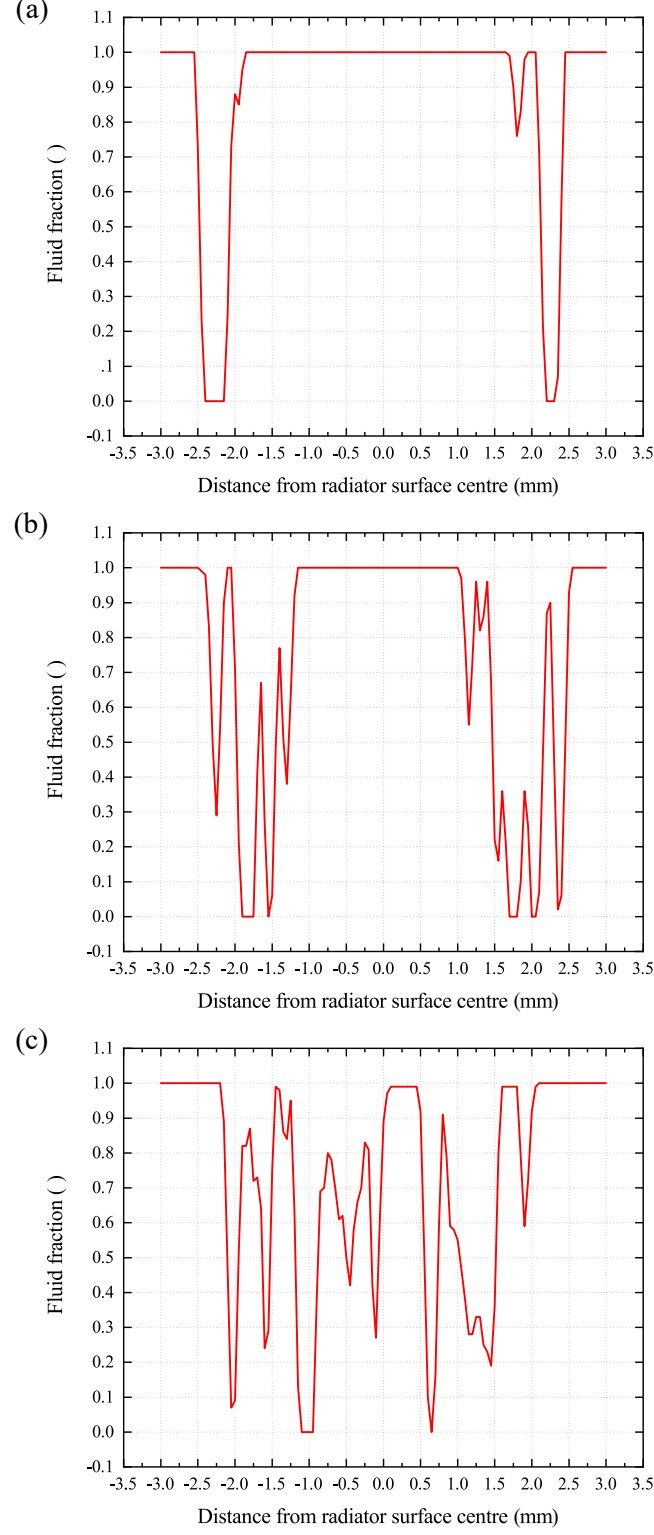

**Figure 7.** Fluid fraction analysis on the radiator surface: (**a**) t = 0.00027 s, (**b**) t = 0.00037 s and (**c**) t = 0.00097 s.

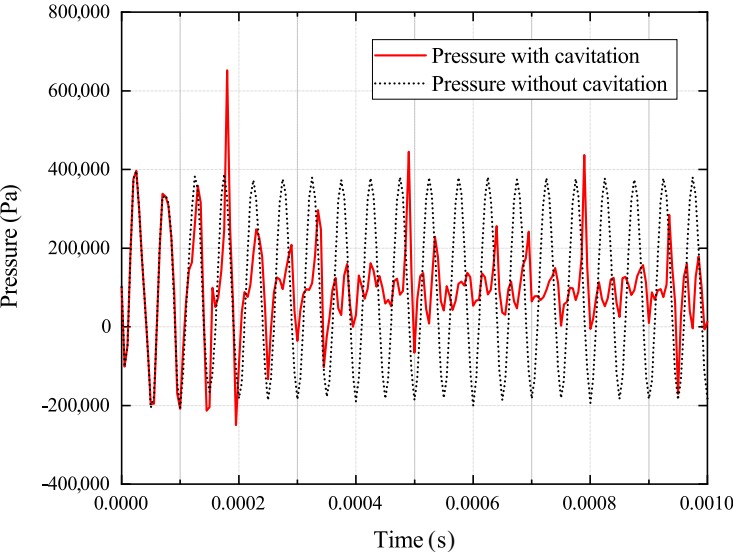

**Figure 8.** Comparison of acoustic pressure amplitudes 4 mm below the radiator with and without the activated cavitation model.

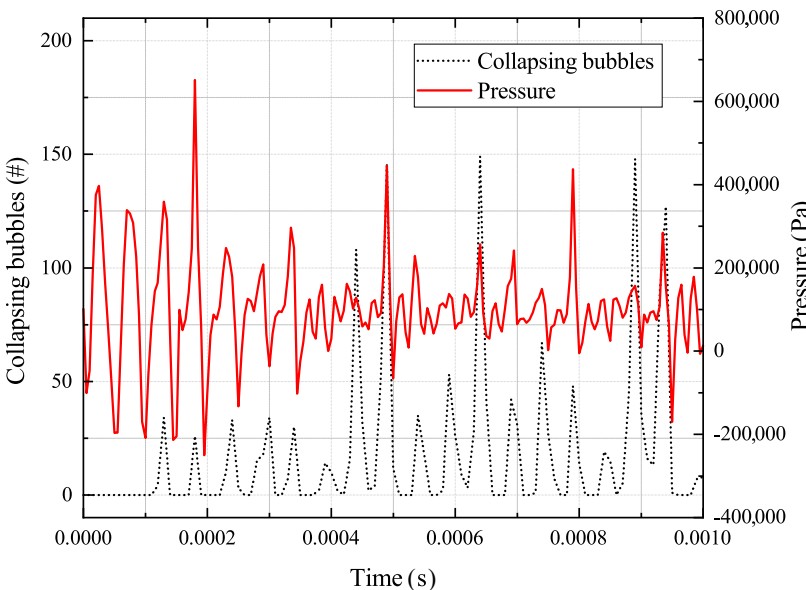

**Figure 9.** Bubble collapse distribution compared to pressure amplitudes.

## 4. Discussion

Based on the results obtained, it can be stated that the software is able to qualitatively simulate the acoustic cavitation effects. The accurate calculation of radiator movement, depending on a sufficient time step setup, was verified, as can be taken from Figure 2. The emergence and propagation of active cavitation areas, including cavitation volume fraction per cell, was calculated. The visualization of cavitation opening and closing depends on and is limited by the chosen cell size. For this reason, not all or just sufficiently sized cavitation bubbles are shown. A smaller cell size would result in a greater amount of opening and closing activity, but would require higher computing times. However, even if the cell size is too coarse and

neglects opening and closing of every single bubble, the cavitation volume fraction indicated all areas of cavitation activity and collapsing bubbles were counted. In this context, the fluid fraction is another useful indicator for characterizing cavitation areas. Analysing the visualized cavitation bubbles, the qualitative accordance of bubble dynamics is recognizable: cavitation bubbles grow for a while and then suddenly contract. Since the pressure rendering allows for the pressure conditions around the bubbles and after their collapse to be analysed, it becomes clear that the pressure close to the bubble areas differs from the rest of the fluid. After the bubble collapse, positive pressure hotspots occur, adequate for the behaviour of shock wave formation after collapse. The amount of collapsed bubbles could therefore function as a quantitative benchmark for different ultrasonic systems and dimensions. The fact that the collapsing activity decreases with increasing distance to the radiator's surface matches the exponential decrease in oscillation energy and qualitative cavitation dynamic according to:

$$A = A_0 e^{-\alpha x} \tag{8}$$

$$I = I_0 e^{-\alpha x} \tag{9}$$

where $A$ and $I$ are the amplitude and intensity of a plane ultrasonic wave, $\alpha$ is the attenuation factor and $x$ is the propagation distance [1].

Next to the fundamental simulation of cavitation dynamics, the influence of cavitation on the propagation of acoustic pressure is also taken into account. Even for a short investigated time frame, the correlation between cavitation development and the damping of acoustic pressures is simulated. Taking the expanding cavities on the radiator's surface into account, it stands to reason that they are one (major) factor of the shielding effect, as already described in literature. If open or void volumes exist on the surface, the radiator is not able to build up pressure in the same way it does at the areas where the radiator is in full contact with the fluid. This is logical, as the acoustic pressure $P_A$ directly depends on the material's density and the speed of sound:

$$P_A = A_0 \rho c \omega \tag{10}$$

where $\omega = 2\pi f$ is the angular frequency [1]. A lower pressure built up at the void contact areas therefore will automatically lead to a lower acoustic pressure distribution.

From a holistic point of view, the software FLOW-3D (accurately) simulates acoustic cavitation and thus could be a helpful tool for the setup of ultrasonic systems and basic investigations (e.g., erosion analytics). Even if there is still a need for further development of the numerical models for the special application of ultrasonic treatment, FLOW-3D still represents a good basis for advanced numerical developments of acoustic cavitation models, e.g., detailed acoustic behaviour in different liquids. Due to the easy adjustment of fluid and system parameters like viscosity, surface tension, as well as frequency and amplitude, cavitation activity for different fluid states and ultrasonic systems can be investigated. Furthermore, the scale and cell size are largely unlimited within the software, thus allowing for very small-scale investigations with corresponding small cell sizes. Another interesting approach could also be so-called two fluid simulations, which are more complex, but allow a more comprehensive definition of another fluid or gas, i.e., hydrogen.

## 5. Conclusions

In this study, the capability of the CFD software FLOW-3D to simulate acoustic cavitation during ultrasonic treatment with a frequency of 20 kHz and a peak-to-peak amplitude of 35 µm within a time frame of 0.001 s was investigated. The results obtained can be summarized as follows:

- The occurrence, propagation and dynamics of cavitation can be simulated.
- The software allows the analysis of pressure conditions in and around the cavitation zone during bubble lifetime, as well as during and after collapse.
- Volume fraction along the bubble-fluid interface can be evaluated.
- Tracking of collapsed bubbles is possible.
- The influence of cavitation on pressure propagation (shielding effect) is taken into account.
- Further investigations should also take the possibility of so-called two fluid simulations into account.

In summary, it could be shown that some of the essential ultrasonic and cavitation effects can be calculated with the software used. The software thus provides a good basis for the further development of numerical, UST-specific models and thus could be a helpful tool for further developments of UST towards industrial application.

**Author Contributions:** Conceptualization, E.R.; data curation, E.R.; formal analysis, E.R.; funding acquisition, E.R. and S.S.; investigation, E.R.; Methodology, E.R.; project administration, E.R. and S.S.; resources, E.R.; software, E.R.; supervision, E.R.; validation, E.R.; visualization, E.R.; writing, original draft, E.R. and N.B.; writing, review and editing, E.R., N.B., and S.S. All authors read and agreed to the published version of the manuscript.

**Funding:** This research was funded by the Investment Bank Saxony Anhalt/European Regional Development Fund project ZS/2018/08/94143.

**Conflicts of Interest:** The authors declare no conflict of interest.

## Abbreviations

The following abbreviation is used in this manuscript:

UST     Ultrasonic treatment

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
