# Peer review of "CFD Simulation Based Investigation of Cavitation Dynamics during High Intensity Ultrasonic Treatment of A356"

_metals, doi:10.3390/met10111529_

Round 1

Reviewer 1 Report

Comments on

CFD-simulation based investigation of cavitation dynamics during high intensity ultrasonic treatment of A356

by

Eric Riedel and Stefan Scharf

 The study is related to the computational fluid dynamic simulation for the phenomena occurring during ultrasonic treatment of aluminum alloy melt. The authors tried to use a software called FLOW-3D to describe the phenomena. Although the present study tried to predict the phenomena during high-intensity ultrasonic irradiation in an aluminum melt without experiments numerically, there is no validation and verification of present model and no detailed model description about the cavitation dynamics, ultrasound propagation, collapse of cavitation, acoustic streaming and etc. Besides, spatially oscillated distribution, which can be considered by the numerical errors, was obtained by the numerical simulation. Therefore, the reviewer cannot accept the manuscript to be published. 

Author Response

Dear Reviewer,

first of all, I would like to thank you for your time spent in reviewing. I totally understand your point of view and for sure validation and verification of simulation results are important for further steps. From my point of view, it is a feasibility study of a CFD simulation software well established in the foundry industry. For this reason, we investigated its potential for calculating the mentioned effects to simplify/reduce the threshold for the simulation of ultrasonic effects with industrial application. The main objective was to show the possibility of the software as solid basis to other researchers with greater expertise in improving numerical models with special application for ultrasonic treatment.

Please accept our sincere regards and take care,

The authors.

Reviewer 2 Report

The article is devoted to the simulation of cavitation under ultrasonic action on the A356 aluminum alloy. A detailed review of the phenomena that occur during ultrasonic processing of the melt is made. The relevance of the topic of the article is not in doubt due to the need for new materials, the development of metallurgy. Simulation of the processes of ultrasonic treatment of melts using modern software will allow us to move on to the physical modeling of these processes. This will make it possible to optimize the processing conditions of metals in order to obtain the best materials. However, the task of the article is only to assess the possibility of simulating these processes using the CFD-simulation tool FLOW-3D using the example of a specific configuration of an ultrasonic unit and a specific metal.
The article is presented sequentially, in detail, well illustrated.

Meanwhile, there are a number of comments and questions.
1. Equations (3.4) - why do they have exactly this form, where does this follow? Maybe you can make a references?

2. Page 2, line 63, 73. Why was such a time step chosen? Why was this cell size chosen? What influences the choice of these parameters?

3. Page 4, lines 91-95. The description of the notation does not correspond to those in equations (2-4). For example, Cp, fcav, Vcav, Pcav (in the equations it is Cc, fc, Vc, pc).

4. Page 4, line 103. The table number (2?) is missed.

5. Page 4, line 103. Why does the gas density in the bubble not change? Why is the pressure in the bubble equal to atmospheric? Do the pressure and density in the bubble not change with its pulsations?

In general, the article contains interesting new results on the possibility of using the CFD-simulation tool FLOW-3D for calculating the cavitation regime of ultrasonic treatment of liquids (in particular, metal melt). I recommend it for publication.

Author Response

Dear Reviewer,

First of all, I would like to thank you for the time you spent to review, and finally, recommend our manuscript. Furthermore, for us, your listed questions and comments are a chance to further improve our manuscript. Thank you for that opportunity. Please allow us to answer your open points sequentially. All changes and new added sentences are highlighted in the manuscript:

  1. Equations (3.4) - why do they have exactly this form, where does this follow? Maybe you can make a references?

Equations 3 and 4 are the FLOW-3D default equations for calculation of cavitation within the fluid. We added sentences to make it clearer and added the reference to the FLOW-3D user manual.

  1. Page 2, line 63, 73. Why was such a time step chosen? Why was this cell size chosen? What influences the choice of these parameters?

Since we already made experiences in the simulation of ultrasonic treatment with FLOW-3D on a larger scale, we knew that we had to find a compromise between calculation time (depending on cell size, time step definition and CPU power) and visual resolution (depending on cell size). To allow for a high resolution in visual results we decided for a very small scale with a corresponding cell size. In the same way we selected the time step size in such a way, that the sinus curves for moving radiator calculated accurately. That was our minimum requirement.

We tried to make this clearer.

  1. Page 4, lines 91-95. The description of the notation does not correspond to those in equations (2-4). For example, Cp, fcav, Vcav, Pcav (in the equations it is Cc, fc, Vc, pc).

Thank you for the note. We corrected it!

  1. Page 4, line 103. The table number (2?) is missed.

Thank you, we corrected it. It was a mistake within table description.

  1. Page 4, line 103. Why does the gas density in the bubble not change? Why is the pressure in the bubble equal to atmospheric? Do the pressure and density in the bubble not change with its pulsations?

Yes, you are absolutely right. The pressure within the bubbles should change with positive and negative pressure amplitudes resulting from the radiator. Since the average value still is the atmospheric pressure, for simplification we used the average value to reduce complexity of the simulation model, since it was a so-called One-Fluid calculation in which you can define a/one density within the bubbles.

We added a more detailed explanation and an additional point in the summary which mentioned the possibilities of two-fluid model.

Please accept our sincere regards and take care,

The authors.

Reviewer 3 Report

Dear Authors,

the article is interesting and clear. The topic you addressed is of high importance both from scientific and technical point of view. I did not find serious mistakes but the following issues should be addressed:

  1. Line 23 and other: Citations [3-9], [14-17], [16-21], [23-28] are too general. I suggest to extend the information with the short statement what was presented in each cited source.
  2. Line 32: H2O – now it looks like H20.
  3. Line 66: what was the reason to choose such model volume 8x8x8mm? Especially with the relation to sonotrode dimensions. Please explain.
  4. Line 80: why this temperature was used? Please explain.
  5. What was the source (sources) of the parameters given in Table 1?
  6. Line 103: Table numer?

After you consider my remarks I recommend the article for publishing.

Sincerely,

Reviewer

Author Response

Dear Reviewer,

First of all, I would like to thank you for the time you spent to review, and finally, recommend our manuscript. Furthermore, we see your listed questions and comments as chance to further improve our manuscript. Thank you for that opportunity. Please allow us to answer your open points sequentially. All changes and new added sentences are highlighted yellow:

Line 23 and other: Citations [3-9], [14-17], [16-21], [23-28] are too general. I suggest to extend the information with the short statement what was presented in each cited source.

I understand your point. We didn’t want to create an unnecessary extensive introduction, so we just declared the references dealing with this topic. To follow your recommendation, we reviewed the mentioned references and made more targeted choice of references. The number of references results from the high interest in the mentioned effects investigated under several different circumstances. We hope our measure are satisfactory for you, even if they are minor.

Line 32: H2O – now it looks like H20.

Yes, it was a typing error. Thank you for the hint.

Line 66: what was the reason to choose such model volume 8x8x8mm? Especially with the relation to sonotrode dimensions. Please explain.

Since we already made experiences in the simulation of ultrasonic treatment with FLOW-3D on a larger scale, we knew that we had to find a compromise between calculation time (depending on CPU power and resolution (depending on cell size). To allow for a high resolution in visual results we decided for a very small scale with a corresponding cell size. In the same way we selected time step size in such a way, that the sinus curves for moving radiator calculated accurately. That was our minimum requirement. In addition, especially in case of in-situ experiments, sonotrodes with diameters of just several mm used. We added corresponding references.

We tried to make this clearer.

Line 80: why this temperature was used? Please explain.

A temperature of 973,15 K was selected since, in the frame of an isothermal model, the speed of sound was known for that temperature from Tzanakis et al. [33]. We added this note in the text.

What was the source (sources) of the parameters given in Table 1?

We added the sources, on which the values are based.

Line 103: Table number?

Thank you, we corrected it. It was a mistake within table description.

Please accept our sincere regards and take care,

The authors.

Round 2

Reviewer 1 Report

The same in the previous comments.

Author Response

Dear Reviewer,

For further processing I have to answer to your review again. I still understand your concerns, but I see it as a pure feasibility study. Nevertheless, I will keep your criticism in mind for future work and thank you for your important feedback and the time you invested.

Kind regards